# Horror Manga: Themes and Stylistics of Japanese Horror Comics

**Paolo La Marca**

Department of Humanities, University of Catania, 95124 Catania, Italy; paolo.lamarca@unict.it

**Abstract:** The objective of this contribution is to create a first, ideal mapping of a "category" of manga that has experienced and is still experiencing a very successful season. Although they are generically identified with the term "horror manga" or "horror comics," these manga should be placed within a narrative universe so magmatic as to escape, however, any univocal representation. When we speak of Japanese horror, in fact, we tend to imagine well-defined scenarios and stereotypes, often conveyed by some novels, manga and, perhaps even more so, some films that have bewitched the West, such as *The Ring* (1998) and *Ju-on* (2000). Despite the success in Italy, too, of authors such as Umezu Kazuo (楳図かずお, b. 1936), Hino Hideshi (日野日出志, b. 1946) and Itō Junji (伊藤潤二, b. 1963), knowledge of horror manga is limited to a number of works and authors who represent, however, only a small percentage of a far more polychrome and multifaceted narrative universe. In other words, the tip of an iceberg just waiting to be brought to light. This preliminary contribution is intended to trace a path, thematic/narrative in nature, from which the route of "horror" manga can emerge in a diachronic, dynamic and evolutionary perspective. It goes without saying that, dealing with nearly seventy years of horror comic book publications, it will be impossible to make an exhaustive examination that takes into account all publishing realities, large and small. That is why the field of investigation will be narrowed down and focus exclusively on a specific historical period, from its beginnings in 1958 to the boom of the 1980s, examining the most recurrent themes and stylistic features of this time segment.

**Keywords:** manga studies; horror manga; pop culture; comic history; Japanese manga

## 1. The Birth of Horror Manga

The first necessary clarification concerns the noun "horror" (in Japanese language *horā*, ホラー) and the historical period in which it spread in the archipelago. As early as the late 1960s, the term "horror" began to circulate, with increasing pervasiveness, in the press and in everyday language. Before then, however, quite other terms were used in connection with this genre of fiction, comic books or film productions: words such as "mystery," "terror," and "dread" were used. In Japan, the horror genre has often been framed within two narrative macro-universes governed by two distinct keywords: the first is *kaidan* (怪談), a term that can be translated as "ghost stories, tales of terror"; and the second is an adjective, *kaiki* (怪奇), meaning "mysterious, extraordinary, supernatural." These terms appear, time and again, in manga and magazine titles, as subtitles of some works, or even as titles of some comic book series.

As critic Yonezawa Yoshihiro (米澤嘉博, 1953–2006) points out, the horrific/spectral component is present in many manga as early as the boom of *kashihon'ya* (貸本屋), the bookstores where it was possible to borrow books, magazines and manga at decidedly low and popular prices ([Yonezawa 1996](), p. 55; [Orsi 1998](), pp. 78–79; [Kajii 1977]()). The lending bookstore market, which was already particularly flourishing in the immediate postwar period, developed in parallel with the spread of manga published in magazines, such as *Kage* (影, Shadows, 1956) and *Machi* (街, City, 1957), within which stories related to mystery and the supernatural were already present. It would be necessary to wait until 1958, however, to witness the birth of the first magazine devoted exclusively to the world

of horror, fear and mystery. This was *Kaidan* (怪談), by Tsubame, a publishing house affiliated with the Hibari Shobō group (Figure 1). Published from 1958 to 1968 for a total of 101 volumes, *Kaidan* was presented as an anthology of short stories (*tanpenshū*, 短編集), A5 format, with a solid binding and with an initial cost of 150 yen. In the same year, the publisher Hibari Shobō published *Ōru kaidan* (オール怪談), a sister magazine that, as the title suggests, dealt exclusively with ghost and mystery stories, in a publishing venture that lasted nearly ten years and published 84 volumes (Figure 2). In addition to sharing a very similar editorial approach in some ways, *Kaidan* and *Ōru kaidan* vied for the favor of a number of authors, many of whom were active contributors to both. Popular names include Koga Shin'ichi (古賀新一, 1936–2018), Hama Shinji (浜慎二, b. 1936), Shirato Sanpei (白土三平, 1932–2021), Kojima Gōseki (小島剛夕, 1928–2000), Umezu Kazuo and Ibara Miki (いばら美喜, b. 1928).

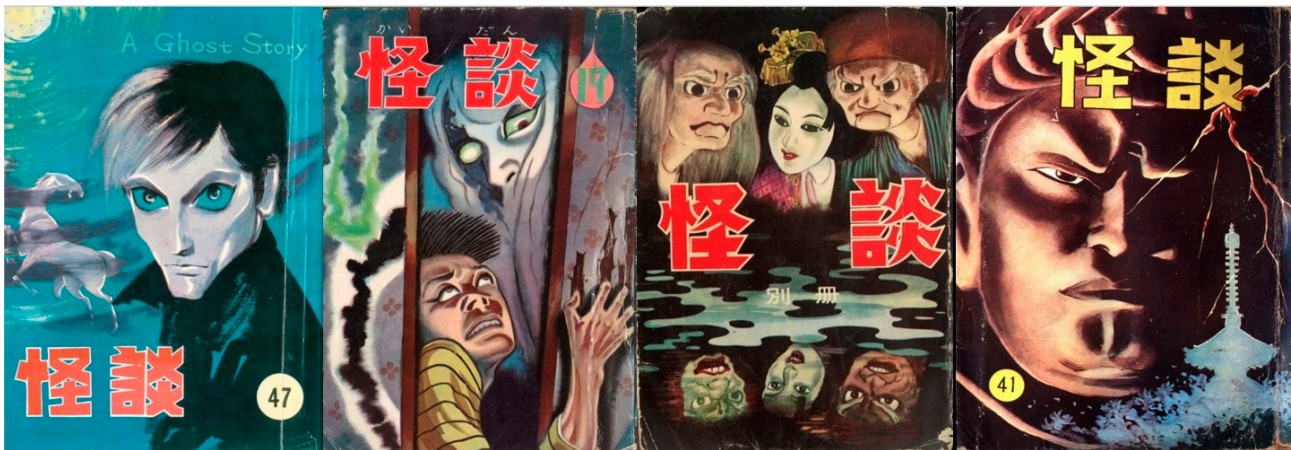

**Figure 1.** Cover of *Kaidan* magazine.

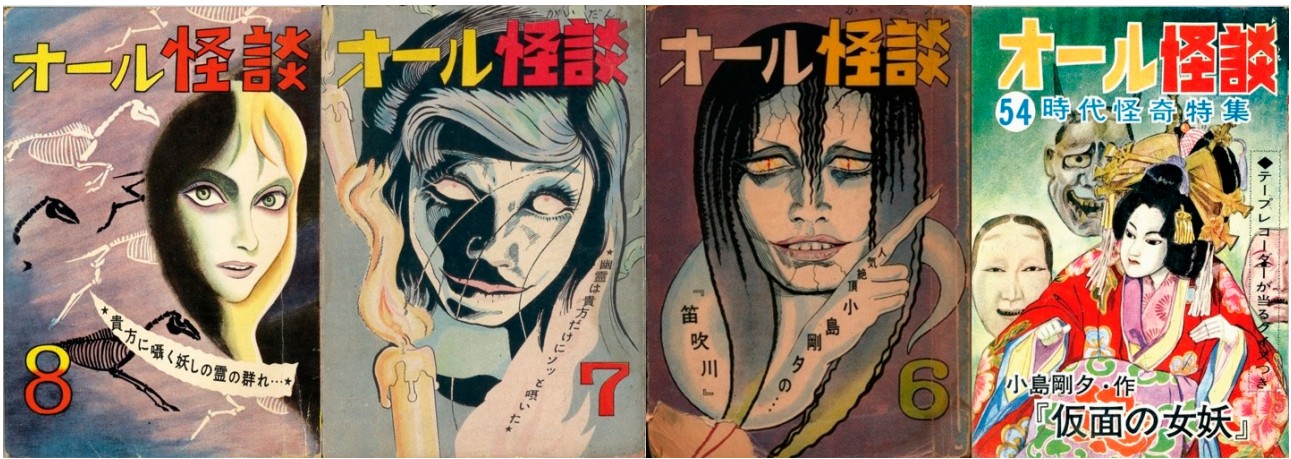

**Figure 2.** Cover of the magazine *Ōru kaidan*.

As a result of the circulation of these two magazines, a growing interest in supernatural-related topics developed, inspired in part not only by the Japanese literary tradition (*kaidanbanashi*, ghost stories) but also by classical Japanese art (woodcut prints with a ghosts and legend theme). The great merit of these two magazines was that they created a format that would later be taken up as a model for the creation of further emulators. Between 1959 and 1960, in fact, many other similar magazines were born that shared a similar basic layout with *Kaidan* and *Ōru kaidan*. These magazines, which are often unavailable today, achieve significant numbers in the comic collector's market, with volumes reaching prices ranging from 5000 to 50,000 yen.

As Yonezawa suggests, the stories presented in these journals can be roughly divided into two macro-groups: on the one hand, stories set in Japan's past (the so-called *jidaimono*, 時代物); on the other hand, those set in the present (the *gendaimono*, 現代物) (Yonezawa 1996, p. 55). In both cases, however, what emerges and seems to unite them is the presence, even in the titles of the stories, of certain *kanji*/synographs that are endlessly repeated: mystery/enigma (*kai*, 怪), strangeness/ordinary (*ki*, 奇) and fear/terror (*kyōfu*, 恐怖). In both the stories set in the Tokugawa era (1603–1868) and those set in the Tokyo of skyscrapers, amidst murders and eerie demonic presences, a deliberate perturbing effect, linked to a sense of anguish mixed with fear, is clearly evident: severed heads, treacherous cats, ignis fatuus accompanying the entry on the scene of spirits thirsting for revenge, evil little girls, ruthless murderers, etc.

According to Yonezawa, the boom in horror manga can also be partly explained by a fortunate combination of publishing/cinematic events. On the one hand, in 1959, the success of an initiative of the Sōgensha publishing house titled *Kaiki shōsetsu zenshū* (怪奇小説全集, Complete Collection of Mystery Novels) was realized; on the other hand, horror films by Hammer Film Production, a British film production company, also began to circulate and gain a fair amount of popularity (Yonezawa 1996, p. 55). The exploit of horror comics should therefore be framed and studied within that precise historical moment and socio-cultural context.

In analyzing the works published in these comic magazines, a summary classification could be attempted, taking into account the main narrative strands: (1) *kaidanmono* (怪談物), stories set in Japan's past, with original tales or simple retellings of ancient ghost legends; (2) *ingamono* (因果物), unfortunate stories linked to a tragic karmic fate; (3) *onryōtan* (怨霊譚), stories of revenge and resentment within which a spirit or ghost returns among the living to avenge a wrong suffered (Yonezawa 1996, p. 55). Over the years, the themes of these comics began to diversify more and incorporate new suggestions from foreign fiction as well. Classic themes of revenge were joined by stories based on irrational or psychological fears, supernatural phenomena, voodoo rituals, magic and witchcraft. Also, since 1960, new publications with "horrific" themes, such as *Kaidan book* (怪談ブック), continued to appear, aiming to build reader loyalty with serialized works rich in pathos and unusual narrative contexts. Emblematic in this regard is the *Shokku shirīzu* (ショックシリーズ, Shock Series) by Baron Yoshimoto (バロン吉元, b.1940), a celebrated manga author who, in those years, still did not sign his *nom de plume*, but with his proper name, Yoshimoto Tadashi (吉元正) (Figure 3).

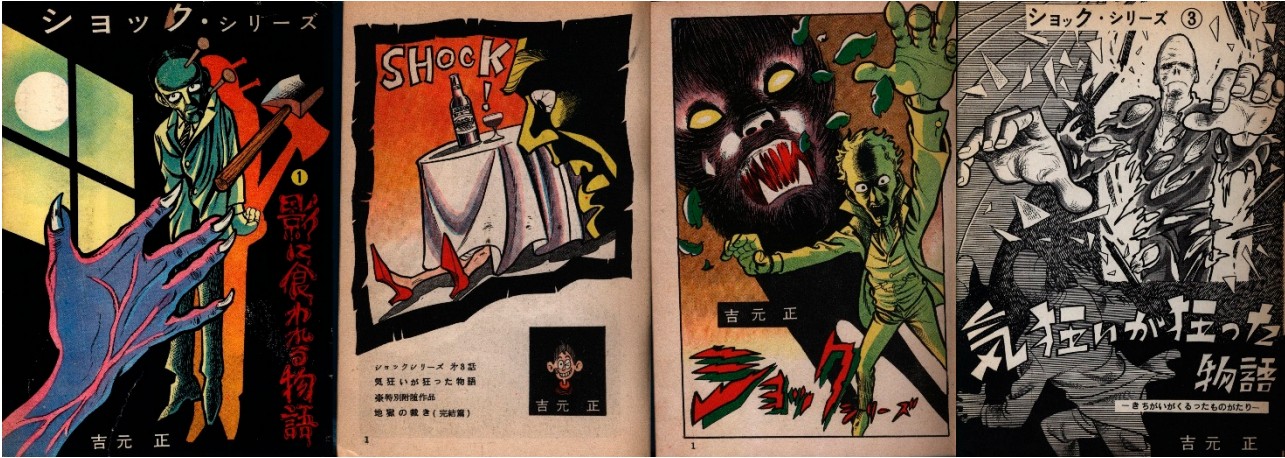

**Figure 3.** Baron Yoshimoto: Shock Series.

Foreign settings (America, France, etc.) influenced subjects of not only an artistic nature (American comics) but also of a thematic nature, including bats (影に食われる物語, *Kage ni kuwareru monogatari*), lizard-men (トカゲ男, *Tokage-otoko*), heinous murders and

Frankenstein-style monstrous beings (気狂いが狂った物語, *Kichigai ga kurutta monogatari*). In addition to Japanese literature, foreign literature, therefore, also began to exert a strong influence on manga writers. In particular, some writers, such as H.P. Lovecraft (1890–1937), enjoyed a fair amount of popularity over the decades, which led them to be repeatedly paid homage to by Japanese comic book writers (Di Fratta 2018, pp. 89–128). Horror imagery, therefore, is enriched with new themes and stylistic features, although, especially in the germinal phase, the stories seem to revolve essentially around four narrative/thematic strands that will be examined below.

### 1.1. Kaibyō (怪猫)

A good number of stories revolve around the vicissitudes of cats endowed with supernatural powers who, through vicissitudes of various kinds, turn into human beings to avenge a wrong suffered (Davisson 2022). Here, too, references to the rich heritage of Japanese folklore, the figures of the *yōkai* (妖怪) and the legends of the past often paid homage to in literature and art as well, are clearly visible. One need only think, for example, of the terrifying cat of the Saga fiefdom (佐賀の怪猫, *Saga no kaibyō*), "the protagonist of an almost inexhaustible epic of felines with supernatural powers" (Orsi 2021, p. V) and the model for an endless series of stories of cats animated by deep resentment and endowed with metamorphic abilities, the so-called *bakeneko* (化け猫). The figure of the cat, understood as a supernatural, vengeful and evil entity, has remained a topos of horror comics, crystallizing one of its most recurring narrative motifs. The cat, often black in color because it is associated, *ça va sans dire*, with night, mystery and all that lurks in the shadows, is an inescapable constant in this genre of stories, declined then in countless variations. One need only think of titles, especially aimed at an audience of young female readers, such as *Kin'iro hitomi* (金色のひとみ, Golden Eyes, 1960) by Kuroda Minoru (黒田みのる, b. 1928), Koga Shin'ichi's *Yami ni hikaru me* (やみに光る目, Eyes Shining in Darkness, 1966), *Neko to watashi to haha to buta* (猫と私と母と豚, The Cat, Me, Mom and the Pig, 1968) by Ikegawa Shinji (池川伸治, 1938–2011) or to *Bakeneko shōjo* (化け猫少女, The Cat Girl, 1982) by Ibara Miki (Figure 4).

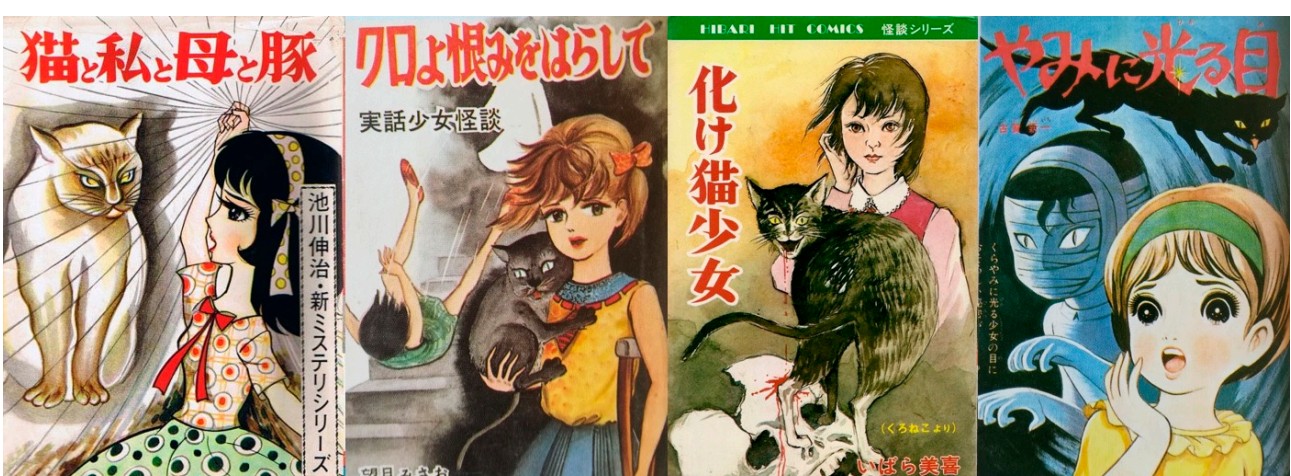

**Figure 4.** *Neko to watashi to haha to buta*; *Kuroyo urami o harashite by Mochizuki Misao*; *Bakeneko shōjo*; *Yami ni hikaru me*.

### 1.2. Henshintan

One of the key topics in many horror manga is the theme of transformation. In addition to the cat, Japanese folklore offers a number of stories of animals such as the fox, snake or crane that suddenly transform into beautiful women either for revenge or to ensnare man and bring him to ruin (Orsi 1988, p. 14; La Marca 2020, p. XIII; Ueda 1988). The figure of the fox-woman, for example, is the pivot around which the plot of *Ōoku no kitsune* (大奥の狐, The Fox of the Ōoku, 1977) by Kamimura Kazuo (上村一夫, 1940–86)

revolves, a story in which lust, seduction and betrayal are mixed with mystery and the supernatural, giving rise to a character with a diabolical dual nature. There are authors such as Kuroda Minoru, for example, who have built a thriving career in the world of *shōjo* manga (少女漫画, girls' comics) precisely because of stories about little girls who turn into animals, or vice versa, animated by deep resentment.

If *bakeneko* and fantastic characters such as the *yuki-onna* (雪女, snow woman) have given rise to stories set in Japan's past, quite different, however, is the approach to the concept of "metamorphosis" for stories set in contemporary times. Often it is the protagonist or main character who undergoes an inexplicable transformation, almost Kafkaesque in memory, into an animal or hybrid/monstrous being.

Transformation becomes the viaticum for a series of irrational fears that disrupt the life of the main character and are masterfully portrayed in the manga of Umezu Kazuo, the undisputed father of horror manga. For Umezu, in fact, one of man's greatest fears consists of not being able to understand himself: not having, therefore, awareness of his own self, the human being perceives himself as another separate entity, a double (Umezu 1996, p. 10). One of the main leitmotifs of Umezu's horror manga rests on a simple fear, namely that one day, suddenly, any human being will stop being himself (or what he mistakenly believed himself to be) and turn into something else. In some of his most famous works, such as *Nekome no shōjo* (ねこ目の少女, The Little Girl with Cat's Eyes, 1965), *Madara no shōjo* (まだらの少女, The spotted girl, 1965) and *Hebi shōjo* (へび少女, The Little Snake Girl, 1966), the protagonists or their antagonists transform into animals or insects, arousing compassion and at the same time revulsion in the eyes of readers (Figure 5). Again, in other Umezu manga, the metamorphosis is intimately connected with the characters' stage of growth and their transition from childhood to adulthood: little girls or boys who suddenly find themselves in an adult body, ill, lacking freedom and autonomy. In this case, horror tries to appeal to the psychological side, feeding unconscious fears related to the body's growth and development.

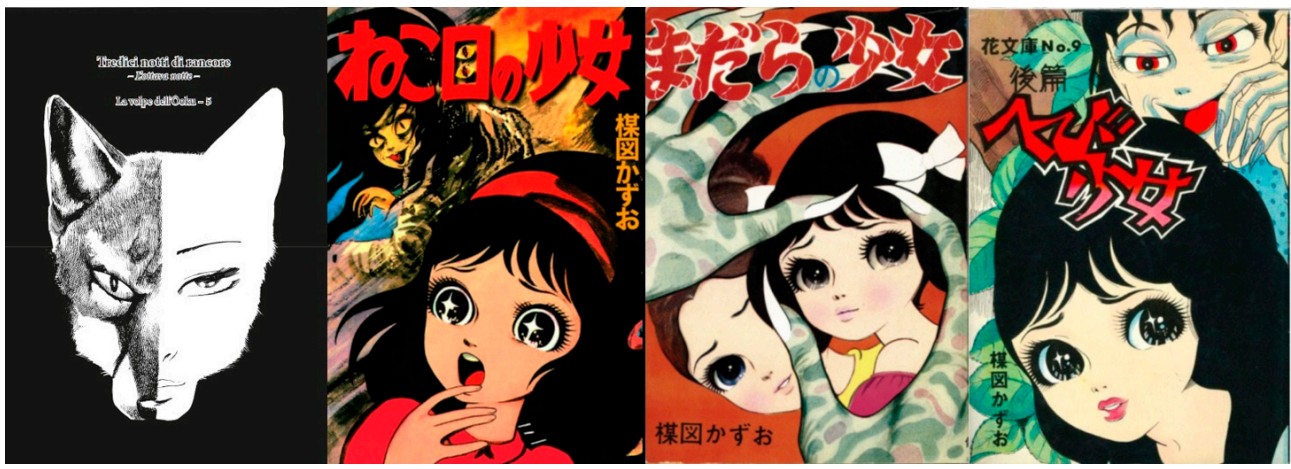

**Figure 5.** *Ōoku no kitsune*; *Nekome no shōjo*; *Madara no shōjo*; *Hebi shōjo*.

### 1.3. Yūrei, Onryō and Revenge

Animating the ranks of horror manga, especially those published between the 1960s and 1970s, is the rich heritage of stories about ghosts, spectres and vengeful spirits. Many of these works draw heavily from the repertoire of *kaidan*, a literary genre centered on ghost stories very much in vogue in the Tokugawa era. Among them, the most popular is undoubtedly *Yotsuya kaidan* (四谷怪談, Ghost Stories in Yotsuya), a famous kabuki theater text written in 1825 by Tsuruya Nanboku IV (1755–1829). The story of Oiwa and Iemon has often been paid homage not only in art by such recognized masters as Katsushika Hokusai (葛飾北斎, 1760–1849) and Utagawa Kuniyoshi (歌川国芳, 1978–1861) but also in cinema in such films as *Shinshaku Yotsuya kaidan* (新釈四谷怪談, 1949) by Kinoshita

Keisuke (木下恵介, 1912–1998) and *Tōkaidō Yotsuya kaidan* (東海道四谷怪談, 1959) by Nakagawa Nobuo (中川信夫, 1905–1984). In the field of manga, there have been countless transpositions, and inevitable retellings, of this story by leading comic book artists such as Mizuki Shigeru (水木しげる, 1922–2015) and Kamimura Kazuo. Between 1976 and 1977, Kamimura made *Onryō jūsan'ya* (怨霊十三夜, Thirteen Nights of Rancor), a work intrinsically linked to *kaidan* fiction and the world of Japanese folklore (Kamimura 2021a, 2021b). The first night, *Nedoshi no Oiwa* (子年のお岩, Oiwa of the Year of the Rat, 1976), is devoted precisely to the *Yotsuya kaidan* and places at the center of the story the figure of an *onryō* (怨霊), that is, a spirit or ghost animated by deep resentment who returns among the living seeking revenge (La Marca 2021). The second night, *Okise no chibusa* (おきせの乳房, The Breasts of Okise, 1976), is also a clear retelling of another famous *kaidanbanashi* (怪談話, ghost story) by San'yūtei Enchō (三遊亭圓長, 1839–1900) titled *Kaidan chibusa enoki* (怪談乳房榎, The Breast-shaped Enoki Tree 1888), itself the subject of countless film and stage adaptations. Kamimura's versions turn out to be all the more interesting precisely because, rather than making it a mere comic book transposition, they show new narrative dynamics without, however, sacrificing the pivotal elements of the original story.

*1.4. Guro*

Among the various "subgenres" of horror comics, the *guro*—from the term *gurotesuku* (グロテスク, grotesque, bizarre, horrific)—still enjoys a certain popularity today. Harkening back to the artistic tradition of *muzan-e* (無残絵), the atrocity prints typical of the Tokugawa era, many authors structured their stories by resorting to crude and gruesome images to shock and frighten the reader: severed heads, blood, disfigured faces, eyes out of their sockets, entrails leaking from the body, torture. Some manga covers replay the top moments of the story, thus introducing the reader to the unreassuring atmosphere of the comic book and announcing to them a good dose of fear. Unlike a movie, however, a manga cannot rely on a soundtrack to punctuate its narrative timing or highlight its moments of greatest emotional intensity. Consider, for example, the shower scene in Alfred Hitchcock's (1899–1980) film *Psycho* (1960) and the role that music plays at that precise moment: without the music and without that emotional crescendo, the scene would certainly have had less emotional impact on the audience. That is why full-page or double-page scenes (so-called splash pages) with gruesome images abound in horror manga. These splash pages have the specific task of shocking the reader at the exact moment when he or she turns the page and is suddenly confronted with a terrifying and visually striking scene. Take, as an example, two of the double splash pages from Kamimura's *Onryō jūsan'ya* (Figure 6). These splash pages unexpectedly strike the reader, unsettling and destabilizing the reading experience.

Among the masters of *ero-guro* manga (short for the terms *erochikku*/erotic and *gutoresuku*/grotesque), it is fitting to mention Maruo Suehiro (丸尾末広, b. 1956), defined by critic Thierry Groensteen (b. 1957) as "the De Sade of contemporary manga." Since his beginnings, Maruo has made the grotesque, combined with racy eroticism and a horrific taste, worthy of the most atrocious Grand Guignol show, his signature style. The horror vein and the erotic vein intertwine and give life to images that, as Giorgio Amitrano suggests, consecrate Maruo as "the modern heir of Yoshitoshi" (月岡芳年, 1839–1892), a celebrated engraver and painter of "atrocious scenes" (Amitrano 2018, p. 23). The comic pages of his manga feature alternating horrors, nightmares, vampires, circus troupes and freaks, not to mention eerie settings that draw heavily from both Japanese horror imagery and German expressionist cinema (above all, *The Cabinet of Dr. Caligari*, 1920). Thanks to a drawing style that combines extreme realism with an almost naïve delicacy, Maruo has succeeded in depicting the darkest hells of body and mind, the worst vilenesses and fears of human beings.

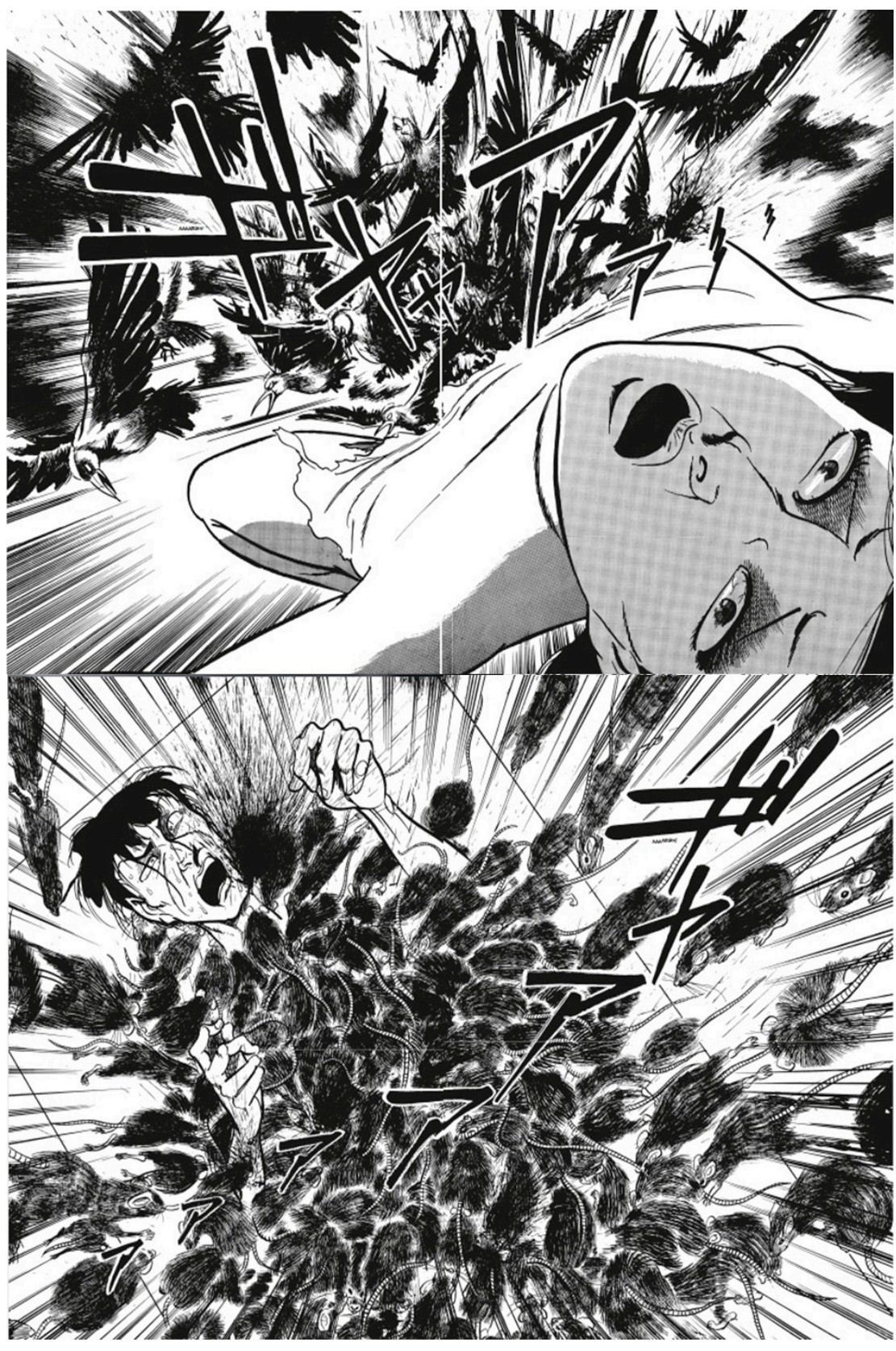

**Figure 6.** Double splash pages from *Okise no chibusa* and *Nedoshi no Oiwa* by Kamimura Kazuo.

## 2. Teenagers and Horror

Beginning in the 1960s, with the birth and boom of weekly magazines, there was an increase in stories related to the world of horror and fear (Yonezawa 1991). An interesting aspect is related to the diversification and layering that horror manga underwent in those years. Labeled as *kyōfu* manga (恐怖漫画, terror manga) or *kowai* manga (こわい漫画, scary manga), these comics began to appeal to a different audience than the adult audience that read magazines such as *Kaidan* and *Ōru kaidan*. This time it is magazines for children and teenagers of both sexes that are publishing, with increasing frequency, stories of terror. Many great authors, such as Mizuki Shigeru, Umezu Kazuo and Watanabe Masako (わたなべまさこ, b. 1929), began in those very years to develop an original artistic path that would lead them to excel in the genre.

Mizuki Shigeru, for example, created the character of Kitarō in the long saga entitled *Hakaba Kitarō* (墓場鬼太郎, Kitarō of the Graveyards, 1959). Within this work, the author has managed to make Japanese tradition and folklore coexist with the entertainment and narrative timing typical of teenage manga. In a world animated by tensions between humans and supernatural beings (the *yōkai*, entities somewhere between monsters and ghosts), Kitarō, a *yōkai* boy, fights to ensure peace and harmony on Earth. In each of his adventures, he is accompanied by strange characters bordering on the grotesque (his father is an eyeball) who help him in his endeavors. Thanks to Mizuki, comics, which in those years were perceived exclusively as a childish entertainment product, began to be seen from a completely different perspective: manga became, therefore, an educational tool as well as a means of rediscovering one's own cultural traditions.

The *yōkai*, as was, moreover, to be expected, arouse the interest of readers to such an extent that, in 1968, another manga was published that re-presented the figure of the man-*yōkai* in the Kitarō style. This was Bem, also the protagonist of a very popular animated series of the same name, even in Italy, entitled *Yōkai ningen Bemu* (妖怪人間ベム, *Bem, the yōkai man*; in Italian, *Bem the human monster)*. The manga, by Tanaka Ken (田中憲, b. 1941), tells the story of three "monsters" (Bero, Bem and Bera) created by a scientist with the task of defeating evil and directing man on the right path, convinced that, sooner or later, they too will be able to abandon their *yōkai* guise and become human beings (Tanaka 2010). The portrait that seems to come out is that of real modern Frankensteins who yearn for humanity and suffer from the hatred and revulsion humans have for them because of their monstrous appearance (Figure 7).

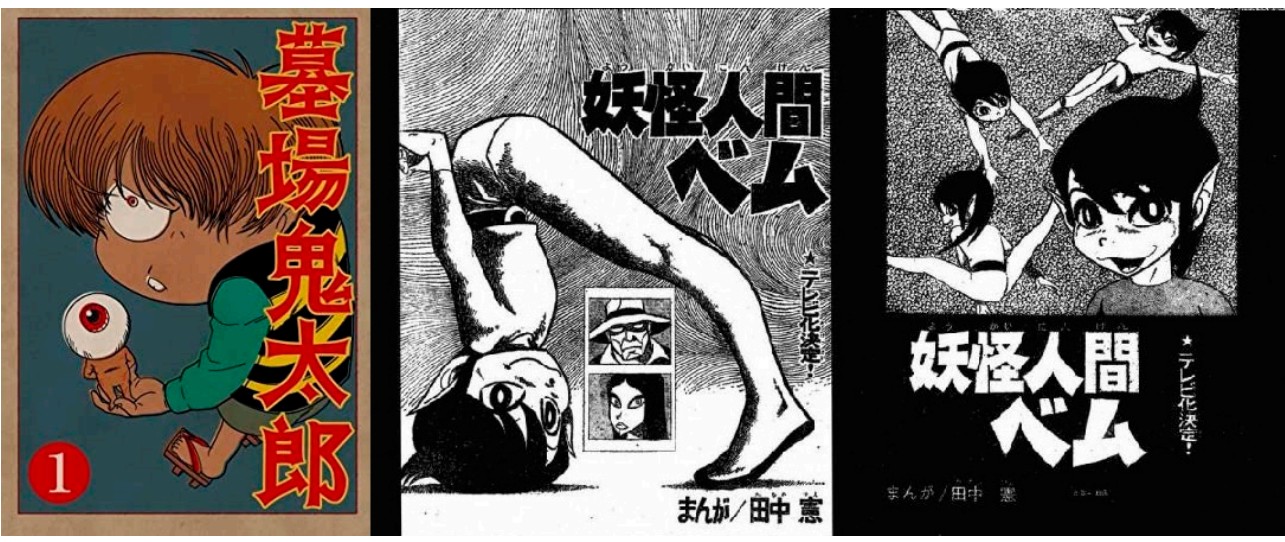

**Figure 7.** *Hakaba Kitarō* and *Yōkai ningen Bemu*.

Remaining in the world of *shōnen* manga (boys' comics), it is impossible not to mention Tsunoda Jirō (つのだじろう, b. 1936), an artist who has linked his name to the horror genre,

inaugurating a long phase of his career (1973–2006) with stories centered on these themes. As an expert connoisseur of the world of the occult, the paranormal and spiritualism, Tsunoda enjoys scattering traces of his studies in his manga, punctually referencing a substantial body of indigenous literature (*Tsurezuregusa*, *Hyaku monogatari* and *Yamato Kaiiki*). Such a choice allowed him to lend credibility to the narrative framework of works such as *Ushiro no Hyakutarō* (うしろの百太郎, Hyakutarō the Guardian Spirit, 1973–76) and *Kyōfu Shinbun* (恐怖新聞, The Daily Terror, 1973–76). *Bōrei gakkyū* (亡霊学級, A Class of Wraiths), published in 1973 in the pages of *Shūkan Shōnen Champion*, is his first work related to the theme of horror and represents his starting point for a disengaged reflection on the world of the occult and mystery. Addressing readers directly, Tsunoda prompts them to doubt the rational world, instilling in their minds the tapeworm of doubt. The incipit of this manga reads:

> *"Hey, I'm talking with you! Have you ever suddenly felt a cold chill down your spine? Of sensing, when you are all alone in your room at night studying, the presence of someone behind you? Of walking alone down a dark street and hearing the sound of someone's footsteps seemingly following you? Of waking up in the middle of a sleepless night and having the feeling of someone pressing down on your chest? […] Yūrei, yōkai, ignis fatuus, an—e--they should not exist! […] Are these really lies? All made-up stories? Yet, there are many people who swear they have seen the spirit of a deceased person"* (Tsunoda 2002, pp. 5–11)

Tsunoda, unlike other cartoonists, does not resort too often to disturbing and gruesome images to shock the reader, but prefers to rely on a simple narrative that can allow for identification between the protagonist and the reader. According to much of the criticism, the absolute verisimilitude of the events narrated in his manga could instill fear in his readers, even by performing trivial everyday actions (waking up in the middle of the night to go to the bathroom; not wanting to open a wall closet with the fear that a ghost might pop out, etc.; Tsuruta 2002, p. 365). For Tsuruta Norio (鶴田法男, n. 1960), the director of the film version of this manga, the "fear" that grips the readers of *Bōrei gakkyū* is of two kinds: "a fear that breaks down the boundary between fiction and reality" and that stages a real feasibility through a fictional story, and "an irrational fear" triggered by an unexpected incident that disrupts the life of a quiet student like so many others (Tsuruta 2002, p. 366). A fear that lurks within us is ready to explode at any moment (Figure 8).

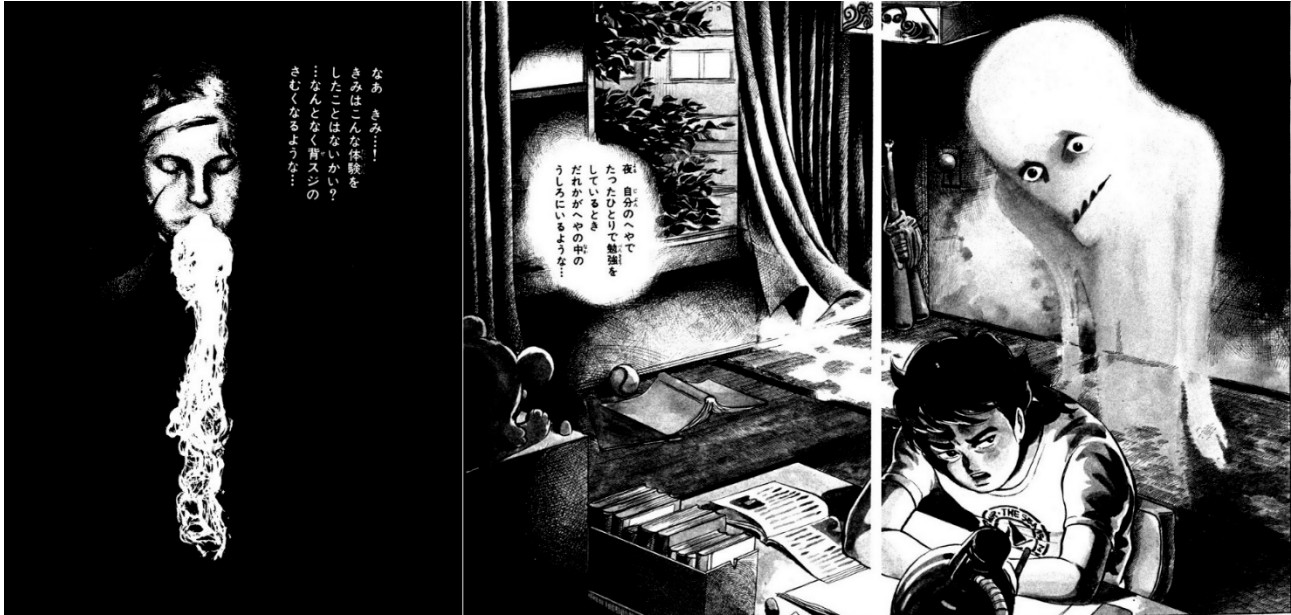

**Figure 8.** Tsunoda Jirō: *Bōrei gakkyū*.

The fear that arises, on the other hand, from Umezu Kazuo's manga is of quite a different nature. Umezu has always sought to "scare" his readers, both male and female, with disturbing stories and images, certainly unusual considering the magazines and the target audience. In an essay of his entitled *Kyōfu e no shōtai* (恐怖への招待, Invitation to Terror, 1988; ed. consulted 1996), Umezu analyzes some aspects of horror comics, identifying recurring iconographic and narratological themes. For the sake of analysis, it was decided to select only two of the aspects that are most relevant to the purposes of this research and that seem to be shared by many horror manga:

*Beauty and ugliness*: Horror manga often and frequently run on a double track characterized by contrasting elements: real and supernatural, youth and old age, beauty and ugliness. In Umezu's manga, this duality is made even more evident through admirable graphic rendering that mixes violence and atrocity with characters drawn as if they were innocent porcelain dolls. The "beauty" of the main characters is contrasted with the "ugliness" of the "antagonists," often victims of society or some botched experiment that makes them unrecognizable or repulsive, as in the manga *Hangyojin* (半魚人, The Fish Man, 1965). In some cases, Umezu plays at reinterpreting the Frankenstein myth by adapting the story in Japanese contexts (small town neighborhoods, classrooms, etc.), effectively making the manga more realistic and, perhaps for that very reason, even more disturbing. In the manga *Senrei* (洗礼, Baptism, 1974), on the other hand, beauty becomes the crux of an actress reminiscent of Gloria Swanson (1899–1983) in her role in *Sunset Boulevard* (Sunset Boulevard, 1950). In order to avoid old age (which in the manga becomes synonymous with "ugliness"), the protagonist is willing to transfer her brain into the body of her beautiful teenage daughter. The maintenance of beauty and the rejection of physical decay, therefore, drive the protagonist to trample on any moral law, desecrating what she holds most precious (La Marca 2014, pp. 66–72). In this way, the manga's comic pages essentially rest on the beauty–brutality pair, first pampering the reader with a few pages in which candor and good feelings dominate, and then unsettling and disturbing him or her with comic pages of rare expressive power in which the dark tones of terror and disquiet dominate.

*Nightmares*: The fear of falling asleep and having nightmares, of not being able to wake up from a nightmare or of finding in dreams a better reality than the one in which one is living, is the central theme of another of Umezu's works, perhaps among the most disturbing of his career, entitled *Kami no hidarite akuma no migite* (神の左手悪魔の右手, The Left Hand of God, the Right Hand of the Devil, 1986). The protagonist is a child who possesses the ability to predict the future in his dreams and, in some cases, the ability to change them (Figure 9). Using ancestral fears, Umezu packs a classic horror film in which he mixes psychological fears and chilling images, offering narrative sequences worthy of the best gore films. Any story could happen anywhere, in the house next door or in the park close to the house, and it is precisely the seeming normality of the everyday that provides Umezu with the key to interpreting and telling the worst fears of children and adolescents. This is because many of the protagonists in his manga are children tested in the face of a terrible truth or victims of a society in which adult men (including parental figures) harbor secrets and murderous instincts.

One of Umezu Kazuo's greatest insights was to identify, in the female target audience, the main consumers of the horror genre. After all, the horror genre has always fascinated female audiences; one need only think of the writer Yoshimoto Banana (吉本ばなな, n. 1964) and her never-concealed fascination with the films of Dario Argento (n. 1940), the novels of Stephen King (n. 1947) and the grotesque manga of Man Gatarō (漫☆画太郎). Umezu Kazuo, then, was one of the first authors to structure *shōjo* manga according to new coordinates, introducing the "fear" variant into a world that seemed immaculate, muffled and safe. Amidst uplifting stories of schoolboy friendships, first heartbeats and fairy tales with a modern flavor, Umezu suddenly broke a narrative code by drawing atrocities and fears. From that moment on, there began to be talk of a veritable boom in *shōjo horā* manga (少女ホラー漫画, shōjo horror manga), stories that, thanks to a strange combina-

tion of "whiteness of the protagonists" and "sense of the macabre/horrific" from the events narrated, captured the attention of thousands of female readers (Umezu 2005). The main names emerging on the comics front are those of Watanabe Masako, Miuchi Suzue (美内すずえ, b. 1951), Takashina Ryoko (高階良子, b. 1946) and Shinohara Chie (篠原千絵, b. 1960). Active in this strand between the 1970s and 1980s, they succeeded in endowing female horror comics with new narrative infrastructures and expressive codes, linking, for example, fear/terror to the concept of growth, physical maturation or, often, to anxieties related to the sexual sphere and relationships with the opposite sex.

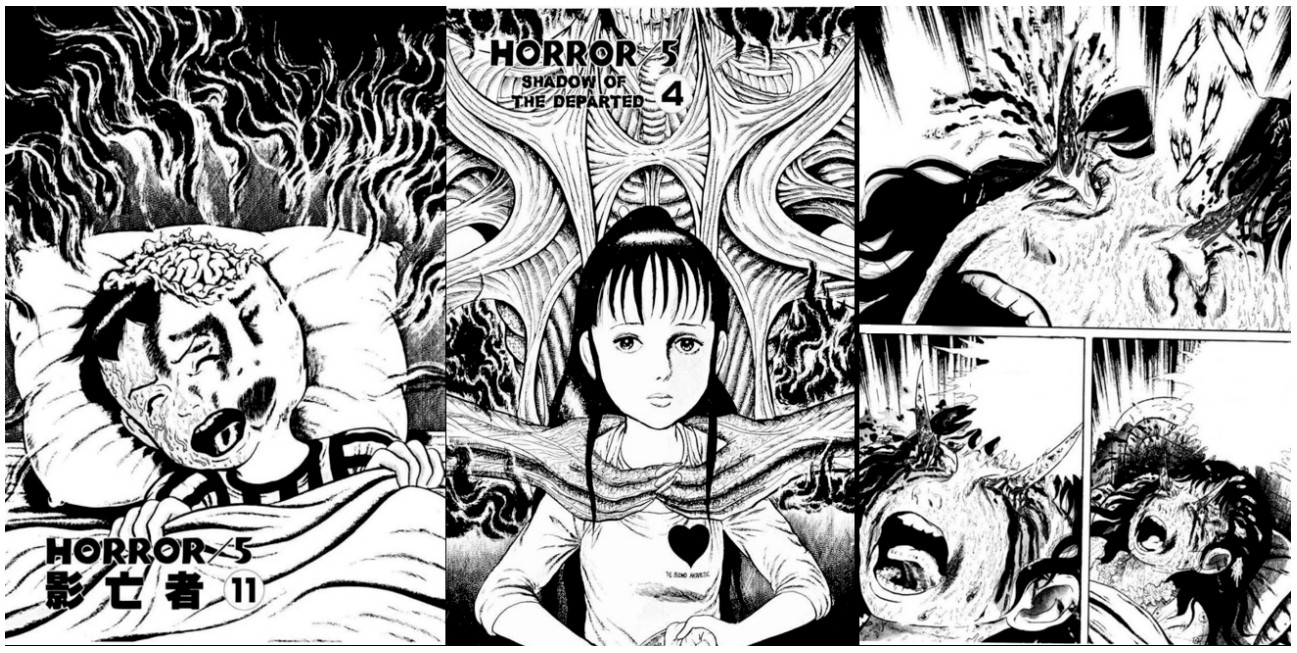

**Figure 9.** Umezu Kazuo: *Kami no hidarite akuma no migite*.

Throughout the 1960s/1970s, different publishing houses continued to present horror titles in mainstream magazines but without devoting a specific or particularly significant publication to them. Horror, understood as a genre, was being given a "quota" in magazines, alongside other narrative genres such as school comedy, sports manga, etc. Just think of titles such as *Debiruman* (デビルマン, Devilman, 1972) by Nagai Gō (永井豪, b. 1945), published in the pages of *Shūkan Shōnen Magazine*. Outside the circuits of the big publishers, however, there were small publishing houses engaged in the creation and dissemination of horror manga, published directly in monographs. The rosiest and most experimental season of horror manga can be found in the very pages of the various volumes of the publishing houses Hibari Shobō, Rippū Shobō and Akebono Shuppan (Figure 10). The Hibari Hit Comics series, for example, has given many authors the opportunity to express themselves in complete freedom and reach the hands of many readers (Hino 2019a, p. 200). Hino Hideshi is the author who, more than others, manages to establish himself in those years thanks to a style that combines rounded, reassuring and almost childlike graphics with a narrative compartment imbued with disarming violence, dripping with blood and horror. Alongside Umezu Kazuo and Itō Junji, Hino Hideshi is regarded as one of the most iconic and controversial horror authors, beloved and translated into several languages (Hino 2019b). Also deserving of rediscovery, however, are the other authors who have also published for Hibari Comics and other publishers and who, although extremely talented, are almost completely unknown outside the borders of Japan. I am referring to authors such as Kawashima Norikazu (川島のりかず), Sugito Kōji (杉戸光史, 1942–1989), Saga Miyuki (さがみゆき, b. 1940) and Mori Yukiko (森由岐子).

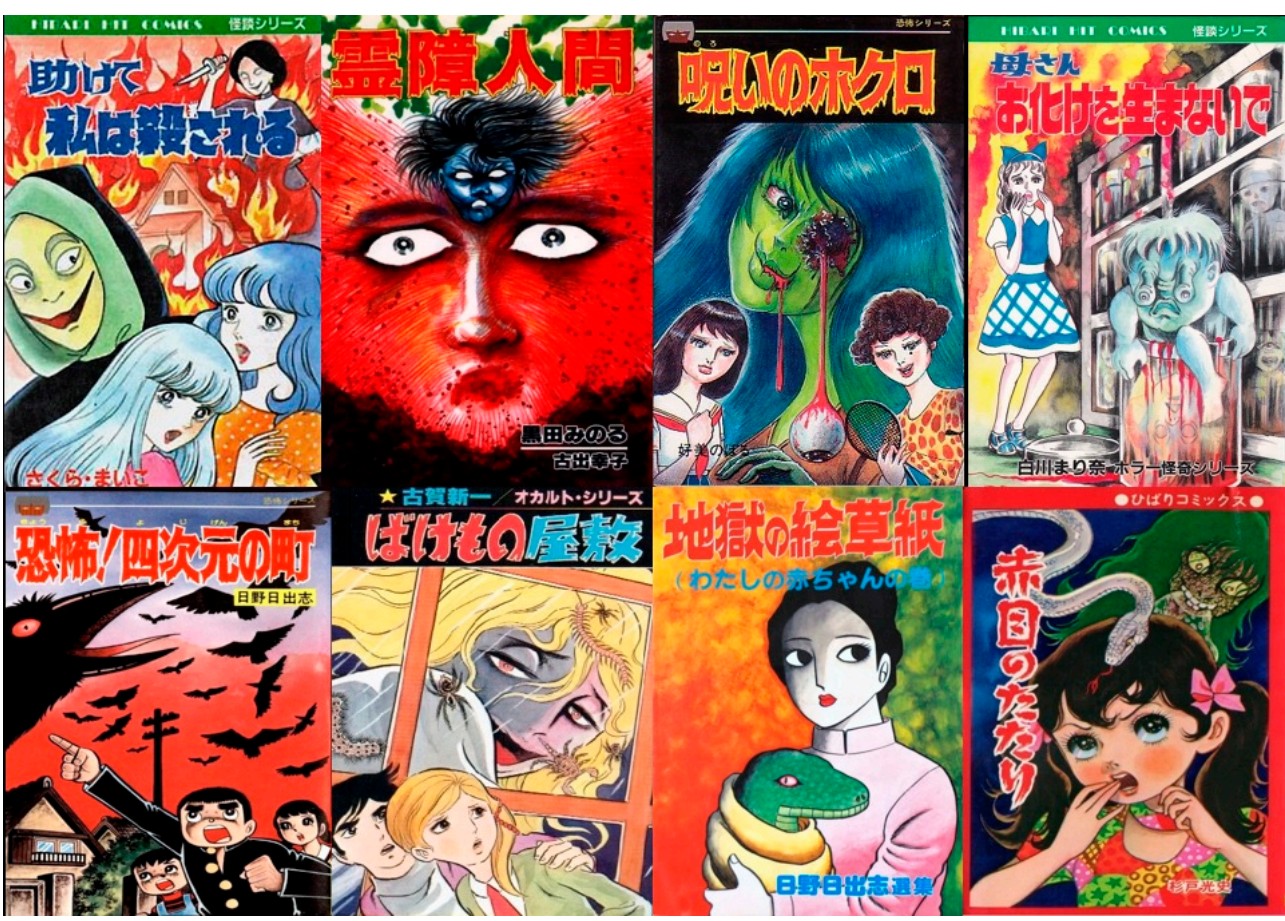

**Figure 10.** Horror volumes, some of them in the Hibari Hit Comic series.

The baton of Hibari Comics would later be picked up in the late 1980s by other publishing entities, such as the magazines *Harouin* (ハロウィン, Halloween, 1986–1995) and *Sasuperia* (サスペリア, 1987–2012) and finally, in the 1990s, by *Horā M* (ホラーM, 1993–2010). It is worth noting that what revived the horror genre and gave it popularity once again was the female audience (VV.AA 2016): after the first big boom of horror *shōjo* manga at the turn of the 1960s and 1970s, the 1980s brought this narrative genre back into vogue thanks to women's magazines such as the aforementioned *Halloween* and, above all, *Suspiria*, whose title is an obvious reference/homage to the *film* Suspiria (1977) by Dario Argento (b. 1940). Amid stories of vampires, killer clowns, eerie demonic presences and urban legends, the manga published in *Sasuperia* boast the signatures of illustrious authors of the past (Hino Hideshi, Koga Shin'ichi and Tsunoda Jirō), combined with newcomers such as Kakinouchi Narumi (垣野内成美, b. 1962) and Inuki Kanako (犬木加奈子, b. 1958), a histrionic, irreverent and perturbing artist who has linked her name to the illustrations of this magazine's most iconic covers (Figure 11) and who, not surprisingly, has been called the queen of horror manga (Midori no Gosunkugi 2021, p. 238).

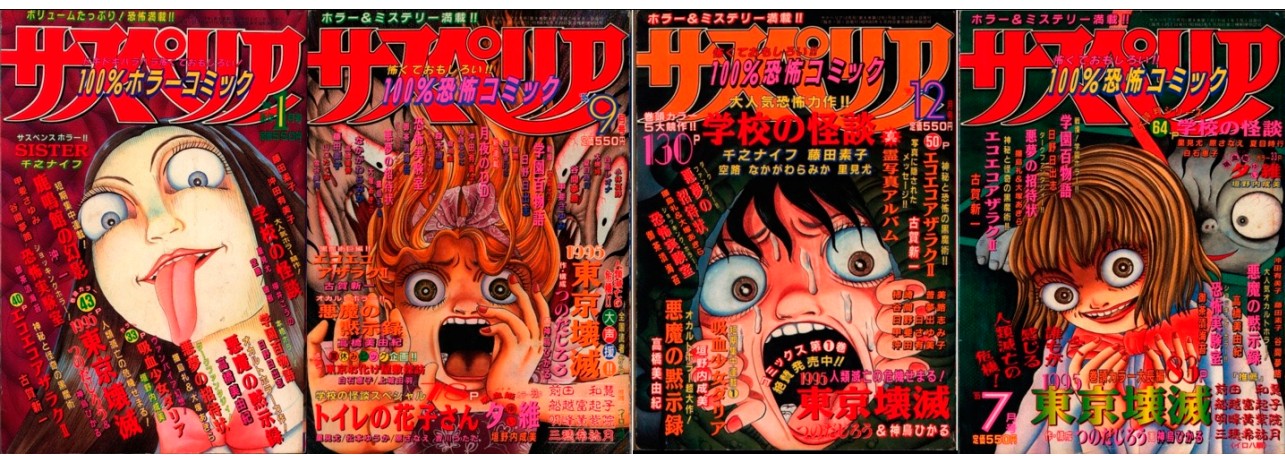

**Figure 11.** Some of the covers made by Inuki Kanako for *Suspiria* magazine.

### 3. Conclusions

This brief examination of Japanese horror comics has illustrated some of the most recurrent themes and stylistic features within this genre of fiction at the turn of the late 1950s and the decade of the 1980s. Developing from the classic ghost-themed storytelling plotlines (the *kaidanbanashi*) of the Tokugawa era, horror comics found an adult-only readership as their first audience. Magazines such as *Kaidan* and *Ōru kaidan*, in fact, are increasingly offering stories poised between the past (ghost tales) and the present (creepy monsters or killers roaming metropolises), thus giving many artists the opportunity to develop a personal sense of aesthetics in relation to themes and drawings. Kojima Gōseki, for example, specialized in historical setting stories, drawing heavily on the rich heritage of creepy/spectral stories in the classical Japanese tradition.

As magazines boomed in the 1960s, there was also an interest in this genre of stories from a younger audience. Although adult magazines continued to feature horror stories, the most notable change related to the emergence of a new readership, namely children and adolescents. These are years in which "terror" seems to have found fertile ground especially among female audiences, to the point of leading to the birth of a veritable publishing strand renamed *shōjo horā* manga. Watanabe Masako, for example, was among the first authors to sense the potential of these stories, scattering references to classic American horror films in her works (above all, *What Happened to Baby Jane?* in *Garasu no shiro*, 1969) (La Marca 2014, pp. 59–62) or telling stories of little girls with angelic beauty who hide, however, a wicked and bloody nature (*Saint Rosalind*, 1973). The general trend of those years is, therefore, to abandon the *kaidan* repertoire in favor of more contemporary and realistic narratives (Japanese schools, mountain villages, city neighborhoods, etc.), so as to facilitate the identification between the reader and the manga protagonist. Only through the plausibility/truthfulness of the narrated events will the reader be able to experience feelings such as anxiety, fear, and terror.

The 1980s represented the golden age of Japanese horror comics (a variety of themes, styles and graphics), thanks to the tireless work of such authors as Hino Hidehsi, Kawashima Norikazu and Saga Miyuki, through the work of publishers such as Hibari Shobō, Rippū Shobō and Akebono Shuppan. This conspicuous legacy was then picked up by magazines aimed at a female audience, such as *Suspiria*, in which an attempt was made to create a synthesis between the "old" and the "new," involving the great artists of the past already active in the days of *Kaidan* and *Ōru kaidan* (such as Koga Shin'ichi) and fostering the talent of new mangaka such as Inuki Kanako.

Today, as never before, the horror genre is experiencing, even in the West, a period of extreme popularity due not only to the rediscovery of the great classics of the past for the first time translated abroad (Umezu, Hino, Itō, etc.) but also to the emergence of works ascribable to new narrative subgenres (zombie, slasher, splatter/gore, survival, etc.)

that are enjoying success and international visibility thanks to targeted cross-media operations (media mix). The key words, however, always remain the same: anxiety, fear, creepiness, terror (Figure 12).

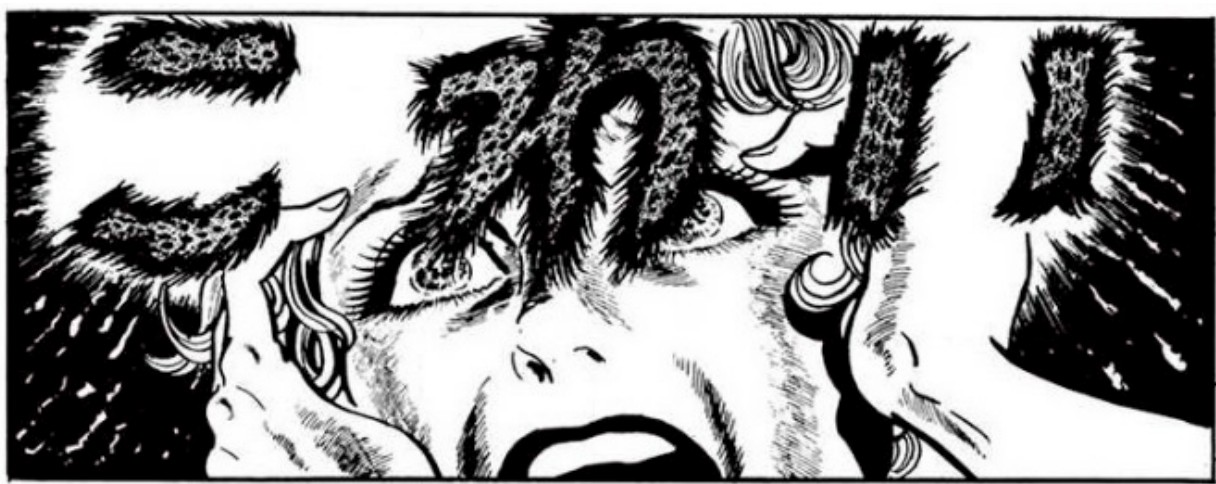

**Figure 12.** Umezu Kazuo: *Orochi*.

**Funding:** This research received no external funding.

**Conflicts of Interest:** The authors declare no conflict of interest.

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
