# Peer review of "Horror Manga: Themes and Stylistics of Japanese Horror Comics"

_humanities, doi:10.3390/h13010008_

Round 1
Reviewer 1 Report
Comments and Suggestions for Authors
There is a misprint in the date of "The Ring", on page 1. I read 1889 instead of 1998.
Author Response
Thank you very much for reviewing my article and for giving a more than positive assessment. Thank you also for pointing out the typo on the date of 'The Ring'.
Reviewer 2 Report
Comments and Suggestions for Authors
The paper shows a fascinating discussion on an intriguing theme, developed with discipline, references variety, well theme-centered dialogue. This study stands as a pillar and fundamental argument over its subject, coming as meaningful to future researchers that might want to broaden or deepen the topic.
La Marca's contribution outlines the manga-horror path from the beginning to the Eighties period, in a dynamic and evolutionary perspective. The reading key is the socio-literary one: manga's history is read by enhancing the exchange relationships between comics, contemporary Japanese literature, and Japanese society changes during the period under examination. Among the perks of this contribution the ability to document the manga-horror universe as not unique and not stereotyped, but polychrome and multifaceted stirkes out.
After delimiting the narrative macro-universe of horror in Japan and its initial diffusion modalities (“according to Yonezawa, the boom in horror manga can also be partly explained by a fortunate combination of publishing/cinematic events”, etc) La Marca gets to the heart of it with a classification that takes into account the main narrative strands and subsequently the suggestions incorporated through the relationship with foreign narrative (as well as with indigenous narrative: see Tsunoda Jirō). Through his analysis progression La Marca highlights both how the stories are enriched with new themes and styles, and how, especially in the initial phase, it is possible to catalogue them essentially around four narrative/thematic strands (examined in succession by the author) by the identification of recurring and/or shared iconographic and narratological themes.
Strong points of this paper:
1. La Marca repeatedly underlines the multiformity, variability, progressive windening of the audience interested in horror-manga: he casts light on how, after a only-adults readers phase, newkind of audience emerged. Infact, on one hand children and teen agers, on the other hand, female readers: this change significantly influenced the production and creativity of authors and authors. Horror manga are becoming more and more meens of discovering oneself, rediscovering one's culture and traditions, coming to represent even an educational tool.
2. The author based his analysis not only on the most iconic and well-known authors work (Umezu Kazuo, Itō Junji, Hino Hideshi), translated into various languages, but also on authors that are completely unknown outside the borders of Japan, despite their great talent (Ka-washima Norikazu, Sugito Kōji, Saga Miyuki, Mori Yukiko). La Marca stresses also out how important the role of some female authors were (from Watanabe Masako to Miuchi Suzu, from Takashina Ryoko to Shinohara Chie, from Ka -kinouchi Narumi to the disturbing and irreverent Inuki Kanako), since they endowed the horror-manga with new narrative infrastructures and expressive codes, linking, for example, fear to the concept of growth, physical maturation or, again, to anxieties related to the sexual sphere and relationships with the opposite sex.
format notes:
Paragraph 1. The sub-paragraphs are all highlighted with the letter A and not with letters in succession.
Paragraph 2. It is marked as 1.
Page 13; Line 10 after images: Sasuperia must be changed to Suspiria
Author Response
I would like to thank you for your thorough analysis of my article on horror manga and for pointing out some typos within the text.
Reviewer 3 Report
Comments and Suggestions for Authors
Although "Horror Manga: Themes and Stylistics of Japanese Horror Comics" contains abundant information about works and authors in the Japanese comic horror genre, it is not ready for publication. The article lacks textual or visual examinations, discussion based on the cultural context, and theoretical references. The author heavily relies on secondary sources, and there is a lack of in-depth original discussion. The section numbering is also confusing, with multiple sections labeled '1' and ‘A’.
To improve, the author should consider narrowing down the topic, improving organization, and more importantly, incorporating deeper examinations and arguments.
Author Response
As you certainly know, manga studies have hardly ever dealt with horror manga in Japan or abroad. Even in the Japanese critical texts I consulted, only brief references are made to the history of horror manga (with the exception of Yonezawa Yoshihiro) and there is no complete and organised examination of it to date. What I have tried to do with this brief article is, as I wrote in the introduction, to create an initial mapping of the genre and to trace a diachronic path on the history of horror manga, from which more reasoned and complex discourses can then be started. The cultural context of this production and the expectations of its readers are sketched briefly when describing the main turning points.
By consulting a rich number of primary sources in Japanese, I believe I produced an original paper, containing information that is not collected in any other previous contribution on the same topic. Nevertheless, taking very seriously your comments on the fact that the paper would rely mainly on secondary sources, if you could indicate to me the secondary sources that you mention, I would be happy to collect and quote them.
Due to copyright issues, it goes without saying that it was not possible for me to undertake 'visual examinations' of the manga that I introduce in the paper. On the other hand, thanks to the generosity of the heirs of some of the mentioned authors, I had the possibility to use for free the images you find in the paper. If "Humanities" is interested in the use of additional materials, I will be happy to provide excerpts of the manga that I could analyze and the contacts to discuss the copyright fees.
Best regards,